# Coagulation Factor XIIIa and Activated Protein C Activate Platelets via GPVI and PAR1

**DOI:** 10.3390/ijms231810203

**Published:** 2022-09-06

**Authors:** Ilaria De Simone, Constance C. F. M. J. Baaten, Martine Jandrot-Perrus, Jonathan M. Gibbins, Hugo ten Cate, Johan W. M. Heemskerk, Chris I. Jones, Paola E. J. van der Meijden

**Affiliations:** 1Department of Biochemistry, Cardiovascular Research Institute Maastricht (CARIM), Maastricht University, 6200 MD Maastricht, The Netherlands; 2School of Biological Sciences, Institute for Metabolic and Cardiovascular Research, University of Reading, Reading RG6 6AS, UK; 3Institute for Molecular Cardiovascular Research, University Hospital Aachen, RWTH Aachen University, 52074 Aachen, Germany; 4UMR_S1148, Laboratory for Vascular Translational Science, INSERM, University Paris Cité, F-75018 Paris, France; 5Thrombosis Expertise Center, Heart and Vascular Center, Maastricht University Medical Center, 6229 HX Maastricht, The Netherlands; 6Synapse Research Institute, 6217 KD Maastricht, The Netherlands

**Keywords:** glycoprotein VI, protease-activated receptor 1, platelet activation, coagulation, coagulation factor XIIIa, activated protein C

## Abstract

Platelet and coagulation activation are highly reciprocal processes driven by multi-molecular interactions. Activated platelets secrete several coagulation factors and expose phosphatidylserine, which supports the activation of coagulation factor proteins. On the other hand, the coagulation cascade generates known ligands for platelet receptors, such as thrombin and fibrin. Coagulation factor (F)Xa, (F)XIIIa and activated protein C (APC) can also bind to platelets, but the functional consequences are unclear. Here, we investigated the effects of the activated (anti)coagulation factors on platelets, other than thrombin. Multicolor flow cytometry and aggregation experiments revealed that the ‘supernatant of (hirudin-treated) coagulated plasma’ (SCP) enhanced CRP-XL-induced platelet responses, i.e., integrin α_IIb_β_3_ activation, P-selectin exposure and aggregate formation. We demonstrated that FXIIIa in combination with APC enhanced platelet activation in solution, and separately immobilized FXIIIa and APC resulted in platelet spreading. Platelet activation by FXIIIa was inhibited by molecular blockade of glycoprotein VI (GPVI) or Syk kinase. In contrast, platelet spreading on immobilized APC was inhibited by PAR1 blockade. Immobilized, but not soluble, FXIIIa and APC also enhanced in vitro adhesion and aggregation under flow. In conclusion, in coagulation, factors other than thrombin or fibrin can induce platelet activation via GPVI and PAR receptors.

## 1. Introduction

Blood platelets and the coagulation system both contribute to hemostasis and thrombosis in a highly interactive manner [1,2]. Well-characterized coagulation products that activate platelets are thrombin and fibrin. Thrombin induces platelet responses via the G-protein coupled receptors, protease-activated receptor (PAR)1 and 4 [3]. Fibrin can stimulate platelet activation, jointly via molecular actions of the integrin α_IIb_β_3_ and glycoprotein VI (GPVI) [4,5], leading to thrombus growth and stabilization [6,7]. GPVI is also known to be the central signaling collagen receptor on platelets [8,9].

Several other activated factors induced by the coagulation process have been described to bind and activate platelets. Factor Xa (FXa) was reported to cleave PAR1 at the thrombin-cleavage site and to evoke platelet responses which were inhibitable by PAR1 inhibitors [10,11]. The formation of FXa occurs at the surface of phosphatidylserine (PS)-exposing platelets by the tenase complex, after which FXa cleaves prothrombin into thrombin in a factor FVa-dependent way [12]. The transglutaminase factor XIIIa (FXIIIa), which crosslinks fibrin fibers, supports platelet spreading and filipodia formation via the activation of Tyr-kinases [13]. In addition, activated protein C (APC, an anticoagulation factor) was found to stimulate platelets via the receptors ApoER2 and GPIb-V-IX [1,14]. Additionally, other factors, such as FV, FIX and FXI, are known to bind to platelets for instance via GPIb-V-IX and integrin α_IIb_β_3_ [15,16,17].

Targeting molecular interaction processes of both platelet and coagulation has been shown to be beneficial in terms of cardiovascular risk reduction. Both, the ATLAS-ACS 2 TIMI 51 study, where the FXa inhibitor rivaroxaban was combined with dual antiplatelet therapy and the COMPASS trial, where a low-dose rivaroxaban administered in addition to aspirin, provided proof that combining platelet and coagulation inhibitors resulted in a lower rate of cardiovascular events, compared to platelet inhibition alone [2,18,19]. In addition, combined antiplatelet and anticoagulation therapy might be promising in other patient populations as well, for example in patients with myocardial injury after non-cardiac surgery (MINS) [20], in which the MANAGE trial recently showed that dabigatran reduced the risk for major vascular complications [21].

In the literature, we encountered a gap in detailed knowledge regarding the relative contribution of key coagulation and anticoagulation factors—other than thrombin and fibrin—in platelet recruitment, platelet activation and thrombus formation. In the present paper, we aimed to close this gap by investigating, on a molecular and signaling level, how the interactions of FXa, FXIIIa and APC with platelets influenced the activation processes of platelets. We hypothesize that all these factors support these processes and hence may act together. We selected FXa, FXIIIa and APC for further investigation because of their central role in the initiation of thrombin generation, fibrin crosslinking and anticoagulation, respectively. In addition, these (anti)coagulation factors are clinically used or are of current high interest for novel therapies to control thrombosis and hemostasis.

## 2. Results

### 2.1. Supernatant of Hirudin-Treated Coagulated Plasma Enhances Platelet Activation

To determine whether activated factors generated during the coagulation process, other than thrombin, affect platelet activation processes, washed platelets were isolated and exposed to supernatant of hirudin-treated coagulated plasma (SCP). SCP was used to mimic the (patho)physiological situation upon injury and activation of extrinsic coagulation. Platelets exposed to SCP were stimulated with varying concentrations of CRP-XL, whereafter platelet activation markers were assessed by flow cytometric analysis (Figure 1). SCP did not induce platelet activation by itself, but significantly enhanced the CRP-XL-induced integrin α_IIb_β_3_ activation (PAC-1 labeling) and P-selectin exposure (anti-P-selectin mAb labeling) by 30–50% over a range of submaximal concentrations (Figure 1A). In addition, the effect of SCP on platelet aggregation in response to CRP-XL, TRAP-6 or ADP was determined using a plate-based aggregation method. Again, SCP significantly increased the percentage of platelet aggregation upon stimulation with submaximal concentrations of CRP-XL, TRAP6 or ADP, compared to noncoagulated (control) plasma (Figure 1B). These results indicated that components generated during coagulation, other than thrombin, support platelet responses.

### 2.2. Effect of Individual Coagulation Factors on Platelet Activation

Since previous literature described that FXa, FXIIIa and APC interact with platelets [10,13,14], we investigated whether these factors contribute to the enhancing effects of SCP on platelet activation. Washed platelets were primed with FXa (10 µg/mL), APC (10 nM), FXIIIa (10 U/mL) and activated with varying concentrations of CRP-XL, thrombin or TRAP-6.

The addition of FXa significantly enhanced the platelet aggregation response triggered by submaximal doses of CRP-XL, TRAP-6 or ADP (Figure 2A). Flow cytometric analysis showed that FXa increased CRP-XL-induced integrin α_IIb_β_3_ activation and P-selectin exposure (Figure 2C). FXa alone also triggered platelet activation (Figure 2B). Strikingly, all effects evoked by FXa were abolished by the thrombin inhibitors dabigatran and hirudin (Figure 2B–D). Consistent with this, increased cytosolic Ca^2+^ levels measured in FXa-treated fura-2-loaded platelets were also inhibited by dabigatran (Figure 2D). This suggests that the FXa-dependent platelet responses are due to the in situ formation of thrombin traces.

In contrast, neither FXIIIa nor APC alone enhanced agonist-induced aggregation (Figure 3A,B). Additionally, platelet activation markers following CRP-XL stimulation were not altered by APC or FXIIIa (Appendix A). However, in line with the effects of SCP, combining APC and FXIIIa significantly enhanced CRP-XL-induced integrin α_IIb_β_3_ activation by approximately 20% (Figure 3C).

### 2.3. Immobilization of APC and FXIIIa Favors Their Activating Effect on Platelets

Since the effects induced by soluble APC and FXIIIa were variable, we examined whether immobilizing APC and FXIIIa could favor their interaction with platelets using platelet spreading assays. FXa was not further investigated, as we observed that effects evoked by FXa were entirely thrombin dependent.

#### 2.3.1. Immobilized APC

Surface-immobilized APC triggered the adhesion and spreading of unstimulated platelets (Figure 4A). Of all adhered platelets, 18.29 ± 13.33% did not undergo shape change, while only a small percentage of platelets protruded filopodia 14.68 ± 9.223% and all others formed lamellipodia 67.03 ± 18.9%. Similar results were obtained for plasma-derived APC (not shown). Since the binding of APC to the EPCR receptor on endothelial cells results in N-terminal PAR1 cleavage [22], we studied a possible role of PAR1 in APC-induced platelet spreading. Therefore, washed platelets were pretreated with the PAR1 inhibitor Atopaxar before spreading on APC-coated surfaces. Atopaxar substantially reduced platelet adhesion by 59.96 ± 15.04% (*p* = 0.0131) to the APC-coated surfaces and abolished the formation of lamellipodia (*p* < 0.001), demonstrating the role of PAR1 in APC-induced platelet spreading (Figure 4A).

#### 2.3.2. Immobilized FXIIIa

Immobilized FXIIIa also induced platelet adhesion and spreading (Figure 4B). The majority of platelets 69.41 ± 15.04% formed lamellipodia on FXIIIa surfaces. To assess the contribution of the transglutaminase activity of FXIIIa in platelet adhesion and spreading, platelets were pretreated with the transglutaminase inhibitor T101. There was no significant decrease in platelet adhesion and in the formation of filopodia and lamellipodia when transglutaminase activity was blocked (Figure 4B).

Since previous studies have demonstrated that the mechanism of FXIIIa-induced platelet spreading and filopodia formation was dependent on integrin α_IIb_β_3_ and tyrosine-kinase activity [13,23,24], we preincubated the platelets with an inhibitor of the kinase Syk, PRT060318 (Syk-IN). Platelet adhesion on surface-immobilized FXIIIa was reduced (*p* = 0.001) and lamellipodia formation was abolished (*p* < 0.0001) after treatment with Syk-IN (Figure 5A). In platelets, the binding of Syk to the β3 cytoplasmic domain of α_IIb_β_3_ integrin is known to be important in lamellipodia formation [25]. However, Syk is also a major signaling molecule downstream of GPVI. Therefore, we reasoned that this may indicate the stimulation of GPVI signaling by FXIIIa. Platelets were therefore treated with the small-molecule GPVI inhibitor honokiol [26] or the blocking anti-GPVI Fab 9O12 [27] with or without the integrin α_IIb_β_3_ inhibitor tirofiban. Treatment with either GPVI inhibitors or tirofiban significantly reduced platelet lamellipodia formation on FXIIIa (*p* < 0.01), but not adhesion (Figure 5B). Combination of either one of the GPVI inhibitors with the integrin α_IIb_β_3_ receptor inhibitor tirofiban, resulted in a significant further decreased platelet adhesion and inhibited lamellipodia formation (Figure 5B). These data indicate the importance of GPVI in platelet activation by FXIIIa, and the synergistic roles of integrin α_IIb_β_3_ and GPVI in the binding and spreading of platelets to immobilized FXIIIa.

### 2.4. Immobilized FXIIIa and APC Enhance Platelet Adhesion under Flow

To obtain more insight into the effect of FXIIIa and APC on thrombus formation, whole blood samples were used for the assessment of thrombus formation under flow using the Maastricht flow chamber [28]. Coagulation factors in physiological conditions can be found soluble in the plasma or immobilized by other ligands or vascular cells. Therefore, APC and FXIIIa were added either directly to the blood or coated on a surface. Surfaces were coated with VWF, since this subendothelial matrix protein only induces weak platelet responses. We reasoned that additional platelet effects evoked by APC or FXIIIa would be rather detected on VWF than on more potent surfaces, such as collagens. To investigate whether soluble FXIIIa and APC enhance thrombus formation on a VWF surface, blood was incubated for 5 min with vehicle or FXIIIa or APC. For experimentation with FXIIIa, citrated blood was used, which was recalcified in the presence of PPACK. For the investigation of APC, blood was taken on hirudin, to avoid any inhibitory effects of PPACK on APC [29]. Blood was perfused over a VWF-coated surface, at an arterial wall shear rate of 1000 s^−1^ or venous wall shear rate of 300 s^−1^. There was no difference in overall platelet deposition or microaggregate formation when blood was incubated with vehicle, soluble FXIIIa (sFXIIIa) or soluble APC (sAPC) (Figure 6A and Figure 7A). Interestingly, at an arterial wall shear (1000 s^−1^), co-coating FXIIIa or APC with VWF significantly enhanced VWF-induced platelet adhesion (*p* = 0.026 and 0.043) (Figure 6B and Figure 7B). Additionally, microaggregate formation was significantly increased upon FXIIIa co-coating (*p* = 0.043) (Figure 6B). Moreover, co-coating FXIIIa and APC together with VWF increased platelet adhesion to the surface with 32% or 15% compared to co-coating APC or FXIIIa with VWF alone, respectively (not shown).

## 3. Discussion

Platelet and coagulation activation occur contemporarily, but are often studied separately, while both mechanisms sustain thrombus formation and impact thrombosis. Our data show that coagulation factors, factors other than thrombin or fibrin, can induce platelet activation. FXIIIa induced platelet spreading via GPVI, and APC induced platelet spreading via PAR1. As GPVI and PAR1 are interesting targets for novel antiplatelet therapy, it is important to identify the ligands for those receptors. Previously, the only coagulation-derived product which was described to activate GPVI was fibrin(ogen), and the activation of PAR1 by APC was only described for endothelial cells. We observed that the effects of individual (anti-)coagulation factors FXIIIa and APC on platelets were bigger when immobilized than when soluble, and that the combined effects of soluble FXIIIa and APC were capable of enhancing platelet activation. In pathological conditions, i.e., at sites of vascular injury or atherothrombosis, the activated factors FXIIIa and APC, both of which are key in regulating the extent of clot formation, are likely to act in a balanced way to prevent, allow and/or restrict the formation of a thrombus.

Given that during acute thrombotic events platelets are exposed to high levels of multiple activated (anti-)coagulation factors, we investigated the effect of the supernatant of hirudin-treated coagulated plasma (SCP) on platelets. Hereby, we have shown that (anti-)coagulation factors formed upon the activation of the extrinsic pathway jointly enhanced agonist-induced platelet activation and aggregation in a thrombin-independent manner. Although the formed fibrin clot was removed from the SCP and a thrombin inhibitor was added to exclude thrombin and fibrin effects, the residual presence of fibrin(ogen) in SCP and thus its influence on platelet activity cannot be completely excluded. To further explore which elements within SCP could be involved in the enhancement of platelet activation, the effects of individual factors of the coagulation cascade on platelets were investigated, by themselves and in combination.

We observed that the addition of FXa caused platelet integrin α_IIb_β_3_ activation, secretion, aggregation and the release of cytosolic Ca^2+^, confirming the effects reported by others. A report by Al-Tamimi and colleagues [11] concluded that platelet activating effects of FXa were mediated via PAR1 as effects were abolished in the presence of the PAR1-inhibitors SCH79797. Accordingly, Petzold and colleagues reported that FXa-mediated effects on platelets were abolished by the FXa inhibitor Rivaroxaban or the PAR1 inhibitor Vorapaxar [10]. However, our data showed that all effects evoked by FXa were abolished upon the addition of thrombin inhibitors, suggesting that the effects of FXa rely on the in situ formation of low levels of thrombin, activating platelets via PAR1. Since platelets’ alpha granules and open canalicular system contain several coagulation factors and co-factors, the generation of thrombin could possibly be explained by the release of traces of prothrombin and factor V/Va by platelets [30].

APC has previously been shown to mediate cytoprotective effects in endothelial cells via PAR1 signaling [14,31,32], which are inhibited by the orthosteric PAR1 inhibitors Vorapaxar and Atopaxar [33]. We revealed that APC-induced platelet spreading is also dependent on PAR-1 and spreading could be inhibited by Atopaxar. Platelet adhesion, however, was not completely abolished upon PAR1 inhibition, suggesting a complementary role of other receptors, for example ApoER2 and GPIbα, which have also been shown to be involved in APC-induced platelet spreading [14]. Compatible with earlier findings by White et al. [14], we could detect an additive effect of APC on platelet adhesion under flow.

The endothelial cell protein C receptor (EPCR) captures and immobilizes APC on the endothelium. Whether this can influence platelet responses in vivo and contribute to platelet adhesion warrants further investigation. The action of APC in this context is uncertain because as well as facilitating platelet adhesion, the binding of APC to platelets could possibly also limit thrombus growth, by localizing the anticoagulant property of the protein C system on the thrombus. To what extent platelet activating and anticoagulant properties of APC influence thrombus formation and growth remains to be established.

FXIIIa has previously been shown to support platelet adhesion and spreading through Syk and integrin α_IIb_β_3_ [13,24]. For the first time, we show that GPVI also has a role in FXIIIa-induced platelet spreading. We observed that the combined inhibition of GPVI and integrin α_IIb_β_3_ almost completely abolished platelet adhesion and spreading, pointing towards synergistic roles of integrin α_IIb_β_3_ and GPVI in the effects of FXIIIa on platelets. Fibrin(ogen) is, to our knowledge, the only coagulation product previously described that induces platelet responses via the platelet receptor GPVI. Complementary to our findings, Moroi M. et al. reported proof that GPVI-dimer selectively binds to FXIII A-subunit [34]. Our study demonstrates the functional consequences of the interaction between FXIIIa and GPVI.

Our results highlight the importance of the immobilization of FXIIIa, suggesting that FXIIIa may need to be captured before eliciting a platelet response, which may be due to a change in FXIIIa conformation [23], or due to the importance of clustering GPVI [35,36]. Under physiological conditions, FXIIIa is captured by fibrin(ogen), and by platelet surface receptors in a growing thrombus [37]. FXIIIa is also exposed on the surface of activated platelets, suggesting that there are amble sources of immobilized FXIIIa in a forming thrombus [38].

In conclusion, our data provide novel evidence that: (i) coagulation products generated upon the activation of the extrinsic pathway support platelet activation, independently of thrombin; (ii) coagulation factor (F)Xa-induced platelet responses rely solely on the in situ formation of thrombin; (iii) immobilized anticoagulation factor APC induces platelet adhesion and spreading, with a role for PAR1; and (iv) immobilized coagulation factor (F)XIIIa induces platelet adhesion and spreading, through GPVI and integrin α_IIb_β_3_. This provides new insights into the molecular processes that drive the interactions between coagulation-generated factors and platelets, and the roles of the platelet receptors PAR1 and GPVI herein. GPVI is a promising target for novel antiplatelet therapy, given its involvement in the collagen-induced pathogenesis of thrombosis, but its minor role in hemostasis [39]. There is, therefore, a need to elucidate the ligands for the GPVI receptor. Similarly, it is clear that the interplay between platelet and coagulation activation needs to be considered if we are to understand the propagation of thrombus, the formation of pathological thrombosis and the efficacy of novel anti-thrombotic therapies.

## 4. Materials and Methods

Human blood was obtained by venipuncture from healthy volunteers, free from antithrombotic medication after written informed consent in accordance with the Declaration of Helsinki. Protocols were reviewed by the local ethics committee. Blood samples were collected into 3.2% trisodium citrate (Vacuette tubes, Greiner Bio-One, Alphen a/d Rijn, The Netherlands). The first 2 mL of blood was discarded to avoid contact activation effects. All subjects had platelet counts within the reference range (150–450 × 10^9^/L), as determined with a Sysmex XP-300 thrombocounter (Sysmex, Cho-ku, Kobe, Japan). The platelets were isolated, washed and resuspended in Hepes buffer pH 7.45 (10 mM Hepes, 136 mM NaCl, 2.7 mM KCl, 2 mM MgCl2, 0.1% glucose and 0.1% BSA), as described earlier [40]. For further details, see Appendix A.

### 4.1. Preparation of Supernatant of Hirudin-Treated Coagulated Plasma (SCP)

Citrate anticoagulated platelet-poor plasma was obtained from blood samples by a double centrifugation at 2200× *g* for 10 min (22 °C, acc. 9, brake 3; Rotina 380 R, Hettich Benelux B.V., Geldermalsen, The Netherlands). As described earlier [41], the collected plasma was recalcified with 16.6 mM CaCl_2_ and activated with 10 pM tissue factor at 37 °C for one hour, resulting in the extrinsic activation of the coagulation cascade. Fibrin clots were manually removed and the fluid remnant was centrifuged at 22,500× *g* for 5 min (22 °C; Hettich EBA 12, Hettich Benelux B.V., Geldermalsen, The Netherlands) to remove remaining fibrin fibers and cell debris. The collected supernatant was post-treated with 10 U/mL hirudin, at room temperature for 10 min, to completely block residual thrombin activity. No additional inhibitors were added. Treated clot supernatants from four healthy donors were pooled and frozen for later experimentation. See Appendix A.

### 4.2. Statistical Analysis

GraphPad Prism 8 software (La Jolla, CA, USA) was used for statistical analysis. Data are presented as mean ± SD. Mean values were compared using an ordinary one- or two-way ANOVA. The Shapiro–Wilk test was used to test for normal distribution of the data. P-values below 0.05 were considered statistically significant in that: * *p* < 0.05, ** *p* < 0.01 and *** *p* < 0.001.

## Figures and Tables

**Figure 1 ijms-23-10203-f001:**
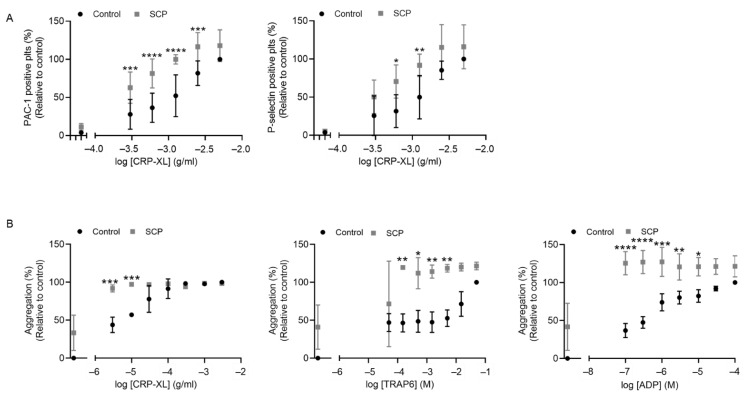
Coagulation-dependent activation of platelets independent of thrombin. (**A**) Washed human platelets in the presence or absence of supernatant of hirudin-treated, coagulated plasma (SCP) were stimulated with a range of CRP-XL concentrations. Activation of integrin α_IIb_β_3_ and P-selectin expression were assessed by flow cytometric analysis, using FITC labeled PAC-1 mAb and Alexa Fluor (AF) 647-labeled anti-human CD62P mAb, respectively. Mean ± SD, *n* = 4. All data were scaled relative to aggregation obtained upon highest CRP-XL concentration in control condition (100%). Ordinary two-way ANOVA, * *p* < 0.05, ** *p* < 0.01, *** *p* < 0.001, **** *p* < 0.0001. (**B**) Aggregation of washed platelets in control or SCP induced by CRP-XL, TRAP-6 or ADP, as assessed by well plate-based aggregation method. Mean ± SD, *n* = 3. All data were scaled relative to aggregation obtained upon highest agonist concentration. in control condition (100%). Ordinary two-way ANOVA, * *p* < 0.05, ** *p* < 0.01, *** *p* < 0.001, **** *p* < 0.0001.

**Figure 2 ijms-23-10203-f002:**
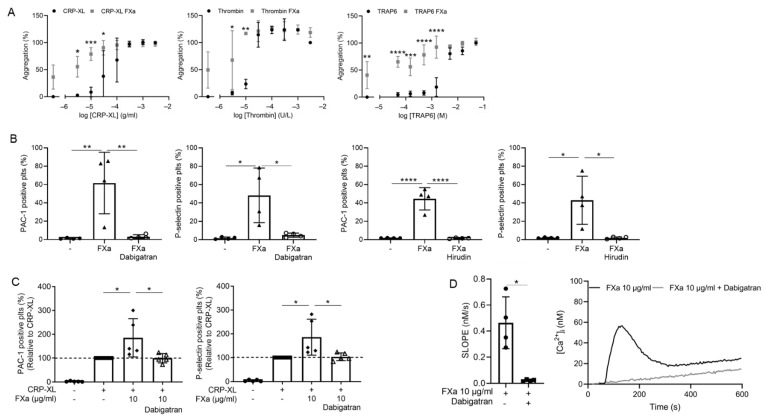
Factor Xa induces platelet activation and enhances agonist-induced platelet effects, in a thrombin-dependent way. (**A**) Factor Xa enhances agonist-induced platelet aggregation. Washed platelets preincubated with vehicle or FXa (10 µg/mL) were stimulated with CRP-XL (0.003–3 µg/mL), thrombin (0.003–3 U/mL) or TRAP-6 (0.05–50 µM). Platelet aggregation was assessed by well plate-based light transmission changes. All data were scaled relative to aggregation obtained upon highest agonist concentration in the presence of vehicle (100%). Mean ± SD, *n* = 3; two-way ANOVA, * *p* < 0.05, ** *p* < 0.01, *** *p* < 0.001, **** *p* < 0.0001. (**B**) FXa-induced platelet activation is inhibitable by dabigatran or hirudin. Flow cytometry, washed platelets. Active integrin and P-selectin expression are shown. Percentage positive platelets. One-way ANOVA, multiple comparisons, mean ± SD, *n* = 4–5, one-way ANOVA, * *p* < 0.05 and ** *p* < 0.01, **** *p* < 0.0001. (**C**) Enhancement of CRP-XL-induced platelet activation by FXa is abolished by dabigatran. One-way ANOVA, multiple comparisons, data are compared to CRP-XL, * *p* < 0.05, mean ± SD, *n* = 4–5. (**D**) FXa-induced cytosolic Ca^2+^ release is inhibitable by dabigatran. Washed platelets, loaded with Fura-2 acetoxymethyl ester (3 μM). Changes in cytosolic [Ca^2+^]_I_ were measured using FlexStation 3. Outcome was assessed from slope of initial Ca^2+^ rises and representative time traces. Unpaired *t*-test, * *p* < 0.05, mean ± SD, *n* = 3–4.

**Figure 3 ijms-23-10203-f003:**
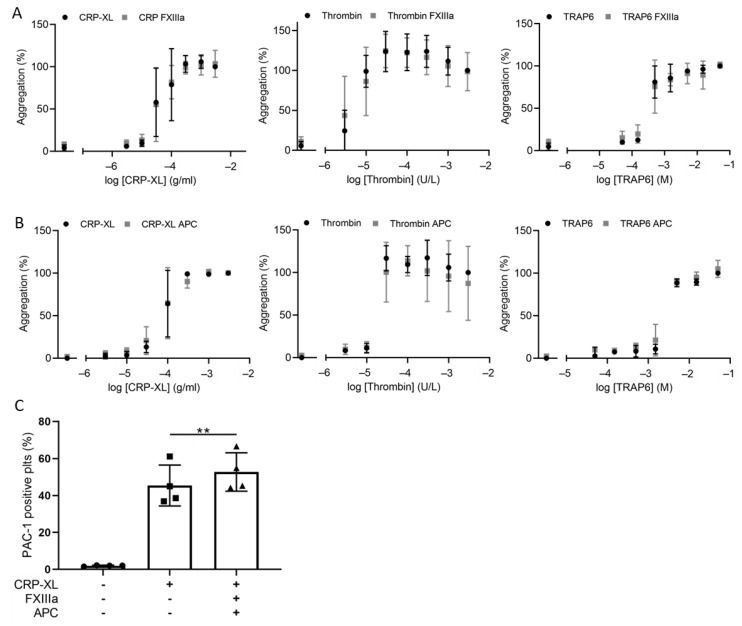
Combinations of coagulation factors (F)XIIIa and anticoagulation factor APC enhance CRP-XL induced platelet activation. (**A**,**B**) Washed platelets preincubated with vehicle, FXIIIa (10 U/mL) or APC (10 nM) were stimulated with CRP-XL (0.003–3 µg/mL), thrombin (0.003–3 U/mL) or TRAP-6 (0.05–50 µM). Platelet aggregation was assessed by well plate-based light transmission changes. All data were scaled relative to aggregation obtained upon highest agonist concentration in the presence of vehicle (100%). Mean ± SD, *n* = 3; two-way ANOVA. (**C**) Washed platelets were preincubated with vehicle or FXIIIa and APC and activated with a submaximal CRP-XL concentration (0.03–0.5 µg/mL). Flow cytometry was used to measure activated integrin α_IIb_β_3_ using FITC labelled PAC-1 mAb. Paired *t*-test, ** *p* < 0.01, mean ± SD, *n* = 4.

**Figure 4 ijms-23-10203-f004:**
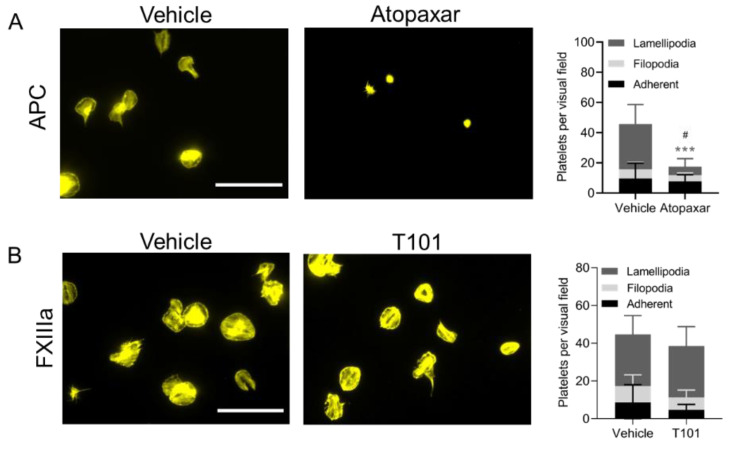
Immobilized APC and FXIIIa induce platelet adhesion and spreading. (**A**) Role of PAR-1 in APC-induced platelet spreading. Washed platelets (20 × 10^9^ platelets/L) were incubated with vehicle (DMSO) or Atopaxar (5 µM) and were allowed to spread for 45 min under static conditions, on a surface coated with APC. The platelets were then fixed, permeabilized and stained with CF543-phalloidin. Spreading was assessed with fluorescence microscopy (bars, 20 µm). Two-way ANOVA, *** *p* < 0.001 compared between lamellipodia, mean ± SD, *n* = 4. Unpaired *t*-test, # *p* < 0.05 comparison of platelets per visual field, mean ± SD, *n* = 4. (**B**) FXIIIa-induced platelet spreading is independent of transglutaminase activity. Washed platelets (20 × 10^9^ platelets/L) were treated with vehicle or transglutaminase inhibitor T101 (20 µM) and were allowed to spread for 45 min under static conditions, on surfaces coated with FXIIIa. Samples were fixed, permeabilized and stained with CF543-phalloidin. Spreading was assessed by fluorescence microscopy (bars, 20 µm). Two-way ANOVA, not significant, mean ± SD, *n* = 3–4. Unpaired *t*-test, not significant, mean ± SD, *n* = 3–4.

**Figure 5 ijms-23-10203-f005:**
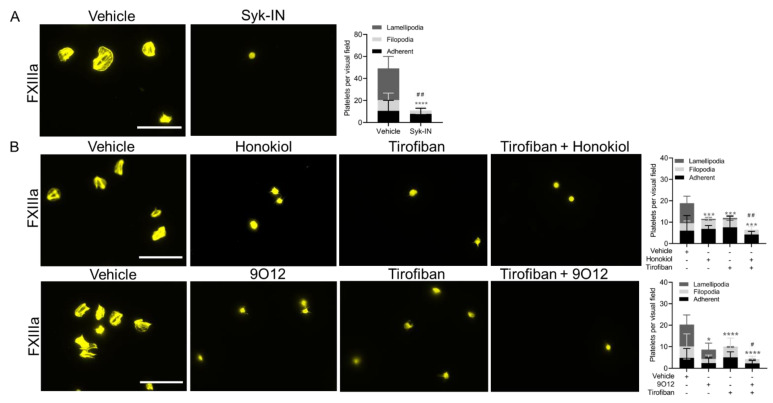
The role of Syk, integrin α_IIb_β_3_ and GPVI in platelet spreading on FXIIIa. (**A**) Spreading on FXIIIa is Syk-dependent. Washed platelets (20 × 10^9^ platelets/L) were treated with vehicle or Syk inhibitor PRT-060318 (20 µM) and added to the FXIIIa-coated surface. Samples were fixed, permeabilized and stained with CF543-phalloidin. Spreading was assessed by fluorescence microscopy (bars, 20 µm). Two-way ANOVA, **** *p* < 0.0001, comparison between lamellipodia, mean ± SD, *n* = 4. Unpaired *t*-test, ## *p* < 0.01, comparison of platelets per visual field, mean ± SD, *n* = 4. (**B**) The role of GPVI in spreading on FXIIIa. Washed platelets (20 × 10^9^ platelets/L) were treated with honokiol (50 µM) or 9O12 (50 µg/mL) ± tirofiban (1 µg/mL) as indicated and added to the FXIIIa-coated surface. Samples were fixed, permeabilized and stained with CF543-phalloidin. Spreading was assessed by fluorescence microscopy (bars, 20 µm). Two-way ANOVA, * *p* < 0.05, *** *p* < 0.001, **** *p* < 0.0001, comparison between lamellipodia, mean ± SD, *n* = 3–5. Unpaired *t*-test, # *p* < 0.05, ## *p* < 0.01, comparison of platelets per visual field, mean ± SD, *n* = 3–5.

**Figure 6 ijms-23-10203-f006:**
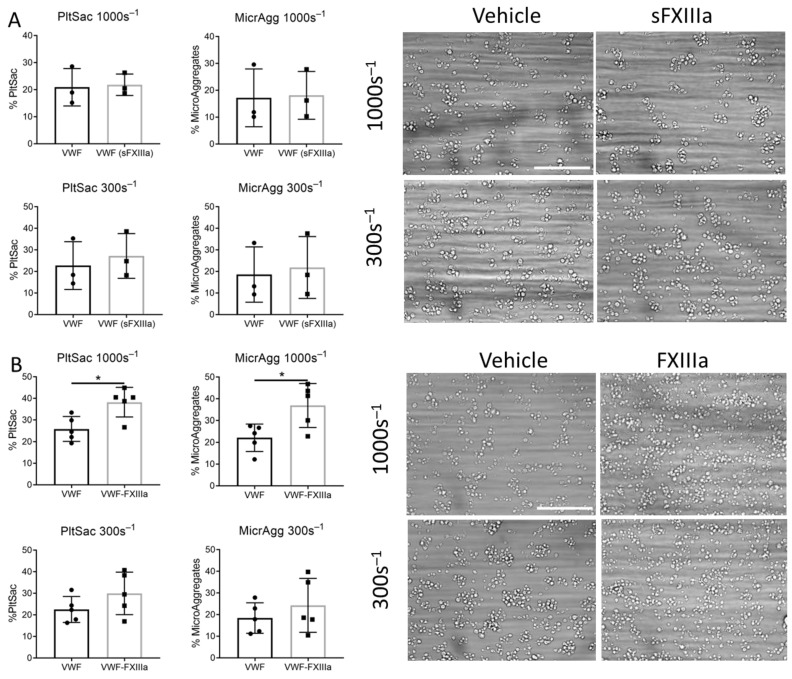
Immobilized FXIIIa enhances platelet adhesion and aggregate formation on VWF. (**A**) Blood samples were preincubated for 5 min with FXIIIa (10 U/mL) and perfused over a surface coated with VWF. (**B**) Blood samples were perfused over surfaces coated with VWF ± FXIIIa. Wall shear rate was 300 s^−1^ or 1000 s^−1^. Representative brightfield images (bars, 40 µm). Platelet adhesion (expressed as percentage surface area coverage, % PltSac). Platelets forming (micro)aggregates (expressed as percentage microaggregates, %MicrAgg). Unpaired *t*-test, * *p* < 0.05, mean ± SD, *n* = 3–5.

**Figure 7 ijms-23-10203-f007:**
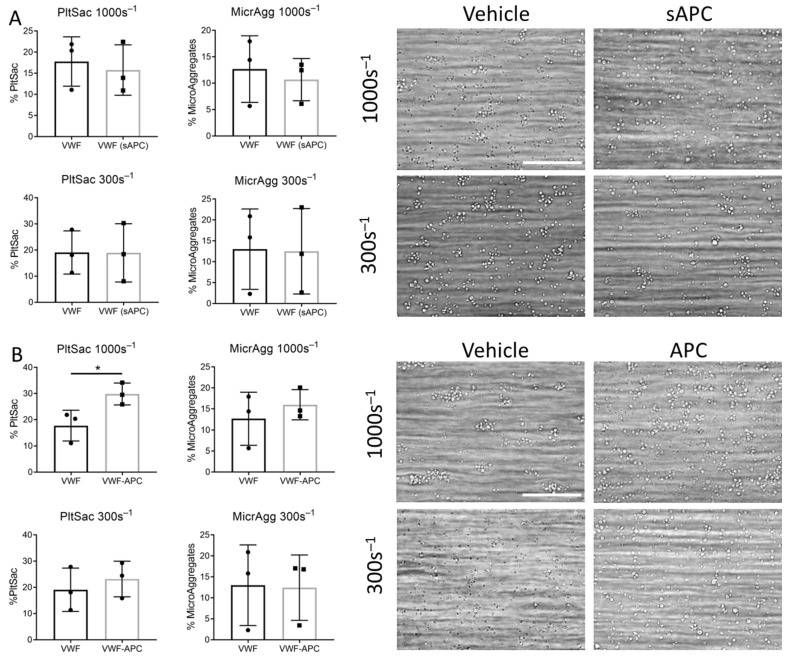
Immobilized APC enhances platelet adhesion on VWF. (**A**) Blood samples were preincubated for 5 min with APC (10 nM) and perfused over a surface coated with VWF. (**B**) Blood samples were perfused over surfaces coated with VWF ± APC. Wall shear rate was 300 s^−1^ or 1000 s^−1^. Representative brightfield images (bars, 40 µm). Platelet adhesion (expressed as percentage surface area coverage, %PltSac). Platelets forming (micro)aggregates (expressed as percentage microaggregates, %MicrAgg). Unpaired *t*-test, * *p* < 0.05, mean ± SD, *n* = 3.

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
