# Peer review of "Coagulation Factor XIIIa and Activated Protein C Activate Platelets via GPVI and PAR1"

_ijms, 2022, doi:10.3390/ijms231810203_

Round 1
Reviewer 1 Report
Review of the Manuscript
ID: ijms-1888257
Type of manuscript: Article
Title: Coagulation factor XIIIa and activated protein C activate platelets via GPVI and PAR1
In this article authors demonstrated that in coagulation, factors other than thrombin or fibrin can induce platelet activation via GPVI and PAR receptors, and thrombus spearing.
Previous literature data described that FXa, FXIIIa and APC interact with platelets and activated them.
In this study laboratory investigation of influence of FXa, FXIIIa and APC on platelet aggregation response triggered by submaximal doses of CPR-XL, TRAP-6 and ADP by using flow cytometry analysis was done.
In this study addition of FXa significantly enhanced platelet aggregation. Furthermore, effects evoked by FXa were abolished by the thrombin inhibitros dabigatran and hirudin, suggesting that they are entirely thrombin dependent. Contrary, neither FXIIIa nor APC alone enhanced agonist-induced aggregation. However, authors observed that effects of individual (anti-) coagulation factors FXIIIa and APC on platelets were bigger when immobilized than soluble. Moreover, study results provide new insight into the molecular processes of interaction between coagulation generated factors and platelets suggesting the GP VI a promising target for novel antiplatelet therapy and better understanding or their role in thrombus propagation.
Laboratory methods and statistical analysis of data used in this investigation are excellently chosen and applicable on different targets for anti-platelet medications.
The most valuable effort of this article is a comprehensive analysis of the literature data on the role of GP VI as well as FXIIIa, focusing on the role of platelets and endothelial cells leading to the thrombus spearing and its clinical consequences.
I strongly recommend the publication of this article as scientific paper.
Reviewer 2 Report
De Simone et al. have investigated whether coagulation factor XIIIa and activated protein C activate platelets via GPVI and PAR1. The authors should be complimented for their extensive efforts in dissecting these mechanisms. I do however have some major and minor points that could be addressed before this article can be considered for publication in the International Journal of Molecular Sciences.
Major points
1. Introduction/abstract
Rationale and relevance
a. A clear rationale for evaluating activated protein C, factor X and factor XIII in one paper is missing. What links those factors other than their potential effects on platelet activation pathways? I fully encourage curiosity driven research; however, your reader must understand why it is important to evaluate the effects of especially these factors on these pathways. I would suggest adding a more on relevance, then go to potential mechanisms (which are also now quite broad) and then rationale to evaluate these factors on platelet activation pathways.
Hypothesis
b. Please also define a clear hypothesis.
2. Methods
Statistical analysis
a. I do not understand why to use mean +/- SD in relatively small numbers. We cannot assume normality within these exploratory numbers. I would recommend to use median with interquartile range, showing all data points or even boxplots with full range.
b. Tests should then be amended to nonparametric tests.
3. Discussion
Here I echo my previous thoughts. Terms as pathological conditions are quite vague. Add why it is important to elucidate the GPVI and PAR1 activation pathways. Then go from there, so your reader can weigh the added knowledge of your paper.
Minor points
4. Methods
Please add a little more on your centrifugation settings: acceleration, brake and temperature and type of centrifuge you used.
5. Results
a. How can you get more than 100% aggregation when you set your scale to 0-100%? (Fig 1 to 3)
b. Just as a clarification, in washed platelet aggregation assay no additional fibrinogen/fibronectin was added?
c. For PAC-1 in flowcytometry, did you also add calciumchloride?
d. Which factors are activated in the supernatant of hirudin treated-coagulated plasma (SCP)? Are there any other factors depleted? This is not clear from your methods/results.
e. I do understand you draw a line to different points of the dose-dependent relationships however this is not longitudinal data, but separately obtained data points. I know many papers do this, but to be very precise we are unsure what happens between two points. I do ask you to address my point on median (IQR) instead of mean (SD) (major points). Also, to analyze them using nonparametric analysis.
However, it is up to you whether you alter the appearance of these graphs, as this is just a suggestion to show them as separate data points and not as longitudinal data.
Overall, I again like to compliment you with these interesting data. I hope to continue our discussion.
Reviewer 3 Report
This is a well conducted study. I have two aspects that should be considered:
1.) Please provide a graphical abstract
2.) Introduction: The authors included ATLAS and COMPASS – please also discuss/consider: DOI: 10.1161/CIR.0000000000001024 and DOI: 10.1016/S0140-6736(18)30832-8
Round 2
Reviewer 2 Report
Dear authors,
Your paper has improved significantly due to the changes you have made. I recommend accepting this paper for publication.
Just as a recommendation I would argue that some of the data are likely not normally distributed (e.g. aggregation data); also for experimental data the same statistical rules apply. I understand that many papers employ normality tests on small (experimental) numbers, but these tests are not designed for n=3/n=4.
Your introduction/discussion including the graph abstract is intelligible and will improve your paper's readability. I enjoyed our discussion and hope you will continue these experiments as they will increase our knowledge of the marvelous ways of platelets.